# Laccase-Carrying Polylactic Acid Electrospun Fibers, Advantages and Limitations in Bio-Oxidation of Amines and Alcohols

**DOI:** 10.3390/jfb14010025

**Published:** 2022-12-31

**Authors:** Valentina Giraldi, Maria Letizia Focarete, Daria Giacomini

**Affiliations:** 1Department of Chemistry “Giacomo Ciamician”, University of Bologna, Via Selmi 2, 40126 Bologna, Italy; 2Health Sciences & Technologies (HST) CIRI, University of Bologna, Via Tolara di Sopra 41/E, 40064 Ozzano dell’Emilia, Italy

**Keywords:** enzyme catalysis, laccases, electrospinning, polylactic acid, oxidation, amines

## Abstract

Laccases are oxidative enzymes that could be good candidates for the functionalization of biopolymers with several applications as biosensors for the determination of bioactive amine and alcohols, for bioremediation of industrial wastewater, and for greener catalysts in oxidation reactions in organic synthesis, especially used for non-phenolic compounds in combination with redox mediators in the so-called Laccase Mediator System (LMS). In this work, we describe the immobilization of Laccase from *Trametes versicolor* (LTv) in poly-L-lactic acid (PLLA) nanofibers and its application in LMS oxidation reactions. The PLLA-LTv catalysts were successfully produced by electrospinning of a water-in-oil emulsion with an optimized method. Different enzyme loadings (1.6, 3.2, and 5.1% ^w^/_w_) were explored, and the obtained mats were thoroughly characterized. The actual amount of the enzyme in the fibers and the eventual enzyme leaching in different solvents were evaluated. Finally, the PLLA-LTv mats were successfully applied as such in the oxidation reaction of catechol, and in the LMS method with TEMPO as mediator in the oxidation of amines with the advantage of easier work-up procedures by the immobilized enzyme. However, the PLLA-LTv failed the oxidation of alcohols with respect to the free enzyme. A tentative explanation was provided.

## 1. Introduction

Laccases (E.C. 1.10.3.2) are oxidative enzymes belonging to the multicopper oxidases family. Thanks to the presence of four copper atoms in the catalytic site, these enzymes can drive the monoelectronic oxidation of four molecules of suitable substrates while one molecule of oxygen is used as terminal oxidant, yielding water as the only co-product [1,2]. For this reason, laccases are potential candidates in the quest for green alternatives to classic oxidants, which often use harmful reagents such as chromium derivatives or are not efficient in terms of atom economy or unwanted by-products [3]. The natural ability of laccases to efficiently oxidize electron-rich aromatic compounds, like anilines or phenols, makes them suitable for a series of biotechnological applications such as the treatment of wastewater, beverage stabilization, delignification of pulp, dyes removal in textile industries, or biosensors for the determination of polyphenolic compounds or biogenic compounds [4]. However, their low redox potential limits their broad application in the oxidation of simple non-aromatic amines or alcohols. To overcome this problem, the laccase mediator system (LMS) was developed: a low molecular weight redox mediator is oxidized by the enzyme and allowed to oxidize different substrates, then the reduced form of the mediator is restored by laccase using O_2_ as the terminal oxidant [5,6]. Several synthetic and natural molecules are used as mediators in LMS, such as 2,2,6,6-tetramethylpiperidin-1-yl)oxyl (TEMPO) and its derivatives, 2,2′-azino-bis(3-ethylbenzothiazoline-6-sulphonic acid (ABTS), hydroxybenzotriazole (HOBt), violuric acid, or some phenolic compounds like syringaldehyde, methylsyringate, and *p*-coumaric acid [7].

The use of laccases, as well as other enzymes, presents some drawbacks related to their limited stability in solution, difficult recovery, and reusability, resulting in high operational costs. Due to the application of laccases in biotechnological processes [4], different immobilization strategies such as adsorption, entrapment, encapsulation, covalent binding, and self-immobilization have been reported, each with their advantages and disadvantages [8,9]. Several supports were explored for laccase immobilization: inorganic materials such as silica, metal oxides, or clays, and organic materials of both natural (alginate, chitosan, agar) or synthetic origin (polyacrylonitrile, polyamides) have been used alone or in combination [10]. Recently, nanostructured materials have gained attention due to their high area-to-volume ratio and easier substrate diffusion [11], and Laccases have been immobilized on nanostructured supports such as mesoporous silica, magnetic nanoparticles, nanotubes, organocomposites [12].

Polymeric nanofibers are suitable supports for enzyme immobilization by covalent binding, absorption, or entrapment of the enzyme inside the fibers directly during the process of the nanofibrous mat production [13]. Electrospinning is a versatile and straightforward technique to obtain nonwoven polymeric fibers with diameters in the micro-nanometric range [14], and Laccases have been immobilized on electrospun scaffolds by adsorption or covalently for applications in biosensors [15], water remediation, or pollutant removal [16,17,18,19]. Dai et al. used poly-D,L-lactic acid (PDLLA) to prepare Laccase encapsulated in polymeric nanofibers by electrospinning homogenous water-in-oil emulsions, to degrade crystal blue dye [20]. More recently, Zdarta et al. used polycaprolactone nanofibers to immobilize laccase by both encapsulation and absorption [21]. The use of different polymers [22] has been reported and additives have been added to the electrospinning solution to enhance special characteristics such as electron transfer processes. Gold nanoparticles and single-walled carbon nanotubes (SWCN) were respectively used in Laccase-poly (vinyl acrylate) composites for catechol detection [23] and in Laccase-poly-lactide-co-glycolide composite for Bisphenol A degradation [24].

Oxidative dehydrogenation reactions are quite useful transformations of functional groups in organic synthesis. In particular, the oxidation of amines or alcohols allows the obtaining of aldehydes or carboxylic acids, respectively [25,26]. However, many of these systems still require transition metals or metal complexes as catalysts, harsh reaction conditions, and create metal-containing wastes. Following our interest in the application of the LMS in organic synthesis [25,26], the present work aims to study the encapsulation of Laccase from *Trametes versicolor* (LTv) in poly-L-lactic acid (PLLA) nanofibers to obtain a recyclable catalyst which can be used for the oxidation of alcohols and amines. PLLA was selected for this study because of its bio-based origin, and for its biocompatibility and biodegradability [27,28].

## 2. Materials and Methods

### 2.1. Materials

Poly L-Lactic Acid (Lacea H.100-E Mw 8.4 × 10^4^ g/mol) was purchased from Mitsui Fine Chemicals (Duesseldorf, Germany) and Pluronic© F127 from Sigma Aldrich, St. Louis, MO, USA. Laccase from *Trametes versicolor* with an activity of 1.49 U/mg, as determined experimentally by catechol assay (see Appendix A), was purchased from Sigma Aldrich. Commercially available reagents and ACS grade solvents were used without further purification. Ultrapure water was obtained by Milli-Q Millipore system. Ibuprofenol 4b was obtained by reduction of Ibuprofen with BH_3._DMS in Et_2_O, while 2-pyridin methanol 4c by reduction of 2-pyridinecarboxaldehyde with NaBH_4_ in MeOH [25].

### 2.2. Fabrication of PLLA-LTv Electrospun Nanofibers

Electrospinning was carried out with a homemade apparatus composed of a high-tension voltage supplier (Spellman SL 50 P10/CE 230, from Spellmann High Voltage Electronics Corporation, Hauppauge, NY, USA), a pump (KD Scientific 200 Series from KD Scientific Inc., Holliston, MA, USA), and a glass syringe connected to a needle (Hamilton NP3-G24, inner diameter of 0.54 mm), through a Teflon tube. The electrospinning apparatus is placed inside a glove box (Iteco Eng., Ravenna, Italy 100 × 7 × 100 cm) equipped with a system to control temperature and humidity. The water-in-oil emulsion of LTv and PLLA for the electrospinning process was obtained as follows: PLLA pellets (300 mg) were dissolved in dichloromethane (DCM, 1.8 mL), then, Pluronic©F127 (33.4 mg) was added, followed by 0.2 mL of dimethylformamide (DMF); separately, different amounts of LTv (5, 10, or 16 mg) were dissolved in 0.20 mL of Milli-Q water to obtain PLLA-LTv fibers characterized by a nominal amount of protein of 1.6, 3.2, and 5.1% ^w^/_w_. The aqueous Laccase solution was then added to the PLLA organic solution and vortexed at 2000 rpm to obtain a stable emulsion. The emulsion was electrospun with the following parameters: applied voltage of 17 kV, flow rate of 23 µL/min, and tip-collector distance of 20 cm. The collector was a circular aluminum foil with a diameter of 10 cm, in a vertical configuration. The obtained mats were stored at 4 °C.

For comparison purposes, blank PLLA mats were also electrospun in the same way without the addition of laccase.

### 2.3. PLLA-LTv Characterization Techniques

Scanning electron microscopy (SEM) imaging of PLLA-LTv samples, after sputter-coating with gold, was performed using an INCAx-sight 7060 apparatus at an accelerating voltage of 15 kV. The diameter of about 100 random fibers was measured on SEM images using ImageJ software. Statistics analysis (average diameter, standard deviation, and diameter distribution) was carried out using Excel.

Differential scanning calorimetry (DSC) measurements were carried out with a TA Instruments Q2000 apparatus. A weighted sample (3–5 mg) from each type of mat was placed inside a Tzero aluminum pan and subjected, under nitrogen flow, to a first heating scan from +20 to +200 °C (scan rate 20 °C/min), followed by quenching at +20 °C; second heating scan up to +200 °C (20 °C/min), followed by slow cooling down to +20 °C at 10 °C/min; and final heating scan up to +200 °C at 20 °C/min. Data were analyzed using the TA Universal Analysis software.

ATR-FTIR spectra were collected on an Alpha FT-IR Bruker spectrometer equipped with a platinum ATR single reflection diamond module. Spectra were recorded with a resolution of 4 cm^−1^ in the 4000–450 cm^−1^ scan range, with 128 scans. As a reference, the background spectrum of air was recorded before the acquisition of each sample spectrum.

Thermogravimetric analyses (TGA) of the electrospun mats were performed with a TA Instrument TGA Q500 apparatus from RT to 700 °C, at 10 °C/min rate, in air.

Water contact angle (WCA) measurements were carried out with a Theta Lite instrument from Biolin Scientific (Alessandria, Italy) and analyzed with OneAttension software. For each sample, seven measurements were done using distilled water. The water drop profiles were recorded from 0 to 10 s while the WCA values were determined at around five seconds.

### 2.4. Quantitation of the LTv Immobilized on PLLA and Determination of Enzyme Activity

The total amount of laccase effectively immobilized in PLLA-LTv mats was determined, after dissolution of the mats, by the Bradford protein assay using the standard addition method [29]. PLLA-LTv mats were dissolved in DCM and extracted with Milli-Q Water (2 × 1 mL). Residual DCM was removed from the collected aqueous layers under reduced pressure. A series of standard solutions were prepared in duplicate by dilution from Laccase Tv mother solutions. In a 96-well plate (Falcon©, Polystyrene Tissue Culture Plate), in duplicate, 60 µL of each standard solution were added from the less to the most concentrated to 60 µL of PLLA-LTv extract. The same process was repeated with a blank PLLA mat as a control. 60 µL of the extract from the blank mat with 60 µL of Milli-Q water was used as the control. Bradford reagent, 120 µL, was then added to each well. Immediately, the plate was shaken for 1 min, using the “shake” function of the plate reader (Thermo Scientific Multiskan EX). The plate was then covered with aluminum foil and absorbance was measured after 20 min at λ = 620 nm.

The enzymatic activity of PLLA-LTv mats was assayed with catechol as substrate, following the increase in absorbance at λ = 405 nm due to its oxidation to benzoquinone (ε = 1260 M^−1^ cm^−1^, see Appendix A for the experimental procedure and calculations).

### 2.5. Release Studies

The release of laccase from the scaffolds was evaluated in the aqueous solutions using the Bradford protein assay. PLLA-LTv samples (approximately 5 mg, 2 × 2 cm) were soaked in 1 mL of the designed solvent (Milli-Q water or acetate buffer 0.1 M at pH = 4.5) and submitted to orbital shaking at 400 rpm. After 2 h, the supernatant was separated from the mat and analyzed by Bradford protein assay. The PLLA-LTv samples were then soaked again with 1 mL of fresh solvent. This procedure was repeated after 24 and 72 h always replacing the supernatant with new solvent. For the Bradford assay, Laccase *Tv* standard solutions were prepared in both Milli-Q water and acetate buffer 0.1 M at pH = 4.5. Then, in a 96-well plate, 125 µL of every standard solution and the blank were dispensed in triplicate. The supernatant of the mat samples was also dispensed as it is. Then, 125 µL of Braford Reagent were added in each well; the solutions were mixed for 1 min, the plate was covered with aluminum foil and the absorbance was measured at λ = 620 nm after 20 min.

### 2.6. Oxidation Reactions

#### 2.6.1. Reaction Monitoring and Analytical Techniques

Merck 60 F254 TLC plates were used in monitoring the reactions.

^1^H NMR spectra were recorded with an INOVA 400 instrument with a 5 mm probe. All chemical shifts were quoted relative to deuterated solvent signals (δ in ppm and J in Hz).

HPLC-MS analysis was carried out with an Agilent Technologies HP1100 instrument, equipped with a ZOBRAX-Eclipse XDB-C8 Agilent Technologies column (flow: 0.4 mL/min; mobile phase: CH_3_CN/H_2_O gradient from 30 to 80% CH_3_CN in 8 min and then 80% CH_3_CN until 25 min), coupled with an Agilent Technologies MSD1100 single-quadrupole mass spectrometer (full-scan mode from *m/z* 50 to 2600; scan time of 0.1 s in positive ion mode, ESI spray voltage of 4500 V, nitrogen gas of 35 psi (1 psi = 6894.7 Pa), drying gas flow of 11.5 mL/min, fragmentor voltage of 20 V).

#### 2.6.2. General Procedure for Amine Oxidation

A PLLA-LTv mat, previously wetted into the reaction solvent, was added to a screw cap vial containing a solution of the amine (0.25 mmol, 1 eq) and TEMPO (0.05 mmol, 0.2 eq) in the appropriate buffer (3 mL). Oxygen was bubbled for 30 s, and the closed vial was then kept under constant shaking at 400 rpm upon an orbital shaker. The reaction was monitored by TLC, bubbling oxygen every time the reaction was exposed to air. When no further improvement in starting material conversion was observed, the reaction mixture was transferred into an extraction funnel, the mat was washed several times with the buffer solution, and the aqueous phase was extracted with EtOAc or DCM (3 × 5 mL). The collected organic layers were dried on Na_2_SO_4,_ filtered and the solvent removed under reduced pressure. For 4-Fluoro benzylamine **1c**, the aqueous phase was extracted with Et_2_O (3 × 5 mL) and the solvent was removed by distillation at ambient pressure. The reaction crude was analyzed by ^1^H NMR and HPLC-MS. The unreacted amine was recovered from the aqueous phase after treatment with NaOH 1 M, followed by extraction with EtOAc (3 × 5 mL), drying on Na_2_SO_4_, filtration, and solvent evaporation.

#### 2.6.3. General Procedure for Oxidation of Alcohols

A PLLA-LTv mat was wetted with the reaction solvent and then added to a screw cap vial containing a solution of the alcohol (0.25 mmol, 1 eq) and TEMPO (0.05 mmol, 0.2 eq) in the appropriate solvent (Milli-Q water or buffer acetate, 3 mL). Oxygen was bubbled for 30 s, and the closed vial was then kept under constant shaking at 400 rpm upon an orbital shaker. The reaction was monitored by TLC, bubbling oxygen every time the reaction was exposed to air. When no further improvement in starting material conversion was observed, the reaction mixture was transferred into an extraction funnel, and the mat was washed several times with Milli-Q water. The aqueous phase was then adjusted to pH 2 by the addition of HCl 1 M and then extracted with EtOAc or DCM (3 × 5 mL). The collected organic layers were dried on Na_2_SO_4_, filtered and the solvent removed under reduced pressure. For substrate 2-pyridin methanol 4c, work-up consisted in lyophilization of the reaction mixture. The reaction crude was then analyzed by ^1^H NMR and HPLC-MS.

## 3. Results and Discussion

### 3.1. Scaffold Fabrication and Characterization

Immobilization of the Laccase *Tv* enzyme within PLLA polymer fibers through the electrospinning technique has some limitations concerning the solubilities of the two species: PLLA is a biodegradable polymer soluble in organic solvents, such as DCM [30], Laccase *Tv*, in turn, is soluble in water or buffer solutions but susceptible to organic solvents that could lead to an eventual denaturation of the protein [31].

Thus, a preliminary study was conducted to find the best conditions to obtain a stable emulsion of PLLA in water-organic solvent mixtures that would generate beads-free fibers (see Appendix A). This was achieved by using a blend of PLLA and Pluronic©F127 in a solution of DMF-DCM (9:1) containing 5% *v/v* of water, at a concentration of 15% *w/v* and 10% ^w^/_w_ for PLLA and Pluronic© F127 respectively. The emulsion was electrospun with the optimized parameters reported in the Materials and Methods section. In the electrospinning of PLLA-LTv, laccase was better solubilized by increasing the quantity of water in the emulsion (9% *^v^/_v_*). We could still form a stable emulsion, and the optimal electrospinning parameters were kept the same as previously determined. Different quantities of lyophilized Laccase Tv were used to obtain PLLA-LTv mats with 1.6, 3.2, and 5.1% ^w^/_w_ of laccase content, respectively named PLLA-LTv-A, PLLA-LTv-B, and PLLA-LTv-C.

The fibers of the three PLLA-LTv mats were analyzed by SEM and the obtained results are presented in Figure 1 and Appendix A. As shown in Figure 1, where the morphology of the blank PLLA-Pluronic©F127 fibers is compared with that of the fibers containing the highest amount of laccase PLLA-LTv-C, regular bead-free fibers were obtained, whose average diameters increased in the presence of laccase, from 0.42 ± 0.16 μm for PLLA-Pluronic©F127to 0.81 ± 0.29 μm for PLLA-LTv-C This result is in accordance with literature data and it could be attributed to an increase in the viscosity of the electrospinning solution [32].

The influence of the laccase content on the thermal properties of the PLLA-LTv mats was studied by TGA and DSC. Appendix A shows the TGA curves of the electrospun mats together with that of plain Laccase *LTv* for the sake of comparison. Laccase *LTv* shows a first broad weight loss step (from room temperature to around 150 °C) ascribable to the evaporation of absorbed water, followed by a two-steps thermal degradation occurring at a temperature of maximum weight loss rate (T_max_) of 320 °C and 500 °C. PLLA-Pluronic©F127 and PLLA-LTv-C showed a very similar thermal degradation behavior characterized by a main weight loss step at T_max_ around 359 °C. It is worth noting that in the TGA curve of PLLA-LTv-C a weight loss of small entity at around 260 °C is observed, not present in the PLLA-Pluronic©F127 mat, that overlaps to the first weight loss step of plain laccase. This weight loss step has been attributed to the presence of the enzyme, even if its precise quantitation was not possible by TGA.

In Figure 2, the DSC curves of the heating scans after quenching from the melt are reported. The thermal behavior of plain PLLA, without the presence of Pluronic©F127, is also depicted (Figure 2, curve A) It is evident from Figure 2 that Pluronic^®^F127 has a plasticizing effect on plain PLLA, decreasing its glass transition temperature (T_g_) from 61 °C to 44 °C and favoring its crystallization during quenching. Indeed, while in the case of plain PLLA, the cold crystallization exothermic peak was followed by an endothermic melting peak of the same entity, in the samples containing Pluronic©F127, the cold crystallization ΔH_c_ is much smaller than the ΔH_m_. The addition of laccase had no significant effect on the thermal properties of PLLA-Pluronic©F127.

Since LMS oxidation reactions are carried out in aqueous solvents, the wettability of PLLA-LTv mats was evaluated. The mats containing Pluronic©F127 can be completely wetted after around 10 min, by soaking them into water or acetate buffer. However, when assessing the hydrophilicity by WCA measurements, there was not a significant difference in the angle values for PLLA-LTv-B, PLLA-Pluronic©F127, and PLLA alone mats (Appendix A and Appendix A on Appendix A), probably due to the short WCA measurement time.

### 3.2. Quantitation of the LTv Immobilized on PLLA and Determination of Enzyme Activity

Bradford protein assay was used to determine the actual amount of the laccase in the fibers [29]. The enzyme was isolated by dissolving PLLA-LTv in DCM and water extraction. In a first attempt, the external standard method was used, using a calibration curve with Laccase *Tv* solutions in Milli-Q water. However, a matrix effect was observed since Pluronic©F127 gave interference with the Bradford reagent, sensible to surfactants [33]. The method of internal standards was used to overcome this problem. Three samples from each type of PLLA-LTv were analyzed, and the results are given as the average value with its standard deviation.

The activity of both free and immobilized laccase *Tv* was evaluated spectrophotometrically, following the increase in absorbance due to the oxidation of catechol to benzoquinone (see Appendix A). Activity assays were performed on three samples from each type of PLLA-LTv mats PLLA-LTv-A, PLLA-LTv-B, and PLLA-LTv-C. Results are given as the Laccase activity expressed in unit per mg of mat; specific activity of immobilized laccase was obtained by dividing the activity of the mat by the actual amounts of loaded laccase previously determined. The results for the three PLLA-LTv mats prepared with different laccase loadings are summarized in Table 1.

As a general observation, the actual amount of laccase determined by Bradford assay resulted lower than the nominal amount. A loss of laccase could occur during the electrospinning process, maybe due to a local instability of the W/O emulsion, which could disrupt the electrospinning jet.

This phenomenon has been observed especially in the preparation of PLLA-LTv-C, where a slightly higher amount of water (0.25 mL instead of 0.2 mL) was necessary to solubilize laccase, thus resulting in a lower immobilization efficiency (34%) with respect to PLLA-LTv-A and PLLA-LTv-B (44 and 54%, respectively).

In PLLA-LTv-A and PLLA-LTv-B mats, the enzyme activity increased on increasing the LTv amount; on the contrary, notwithstanding the highest amount of LTv, PLLA-LTv-C showed an enzyme activity comparable with PLLA-LTv-B. The immobilized laccase retained from 18% to 28% of the free laccase activity after the encapsulation; this loss of activity could be due to the contact with organic solvents (DCM and DMF) during the formation of the emulsion or to the voltage applied during the electrospinning process that could affect the tertiary structure of the enzyme. Moreover, unlike the free laccase, the enzyme is entrapped in the nanofibers, so, from a kinetic point of view, there could be some limitations in the mass-transfer of the substrate and/or product from the active site that could result in a lower catalytic efficiency [34]. This result could be consistent with previous findings that showed how the substrate diffusion rate would be determinant in the catalytic efficiency of diffusion-limited Laccases such as LTv [35]. A tentative to co-immobilize LTv and TEMPO by electrospinning proved unsuccessful, and it produced mats with no activity.

Finally, we explored the ability of PLLA-LTv-B mats to effectively oxidize mmol amounts of catechol. The catechol oxidation was tested at three concentrations 5, 10, and 20 mM (Table 2), following the general procedure for alcohol oxidation reported in the Materials and Methods section. The reaction proceeds in a concentration-dependent manner, and with catechol 20 mM it reached a 64% of conversion in 24 h (entry 1). Considering catechol as a model of phenolic pollutants, which are among the major classes of organic pollutants produced by industrial activities, the PLLA-LTv mat could be applied in biotechnological and industrial processes as well as in bioremediation of phenolics in wastewater [36].

### 3.3. Release of Laccase Tv

Enzyme leaching from the mats was studied in two solvent media, Milli-Q water and acetate buffer pH 4.5 (0.1 M), because these are the solvents for the oxidation reactions [25,26]. The released Laccase from the PLLA-LTv mats was quantified by Bradford assays in the supernatant after two hours of stirring and depicted in Figure 3. Data are reported as release % of LTv normalized to the actual amount of laccase previously determined. All samples present a low degree of LTv release in water, but a greater extent in acetate buffer, probably due to a fiber swelling effect. The release in water was low, 12 and 17% in PLLA-LTv-A and PLLA-LTv-B, respectively, but it increased on the increasing of the LTv content, 40% in PLLA-LTv-C, as if the use of higher amounts of laccase could lead to improper encapsulation or leaching of the overloaded amount of the enzyme on the fiber surface.

Two more refreshes of the aqueous and buffered solutions were studied, one after 24 h and the second after a further 48 h; Bradford assays of the two further refreshes did not give any evidence of the enzyme. This result confirms the absence of LTv leaching after the first 2 h, reinforcing the idea that the initial release was only due to the most superficial enzyme, while encapsulated LTv remains entrapped into fibers.

### 3.4. Application of PLLA-LTv in LMS Oxidation Reactions

To explore the possibility of using PLLA-LTv as a reusable catalyst in oxidation of phenolic or non-phenolic compounds, we focused on the oxidation of catechol as a phenolic model as reported above, and of some selected amines and alcohols. Oxidation of amines to aldehydes or imines and oxidations of primary alcohols to carboxylic acids have been studied using the laccase-mediator system LMS [25,26], and accordingly, we tested the selected substrates with PLLA-LTv mats and TEMPO as mediator.

#### 3.4.1. Oxidation of Amines

*p*-Ome-benzylamine 1a was chosen as a model substrate for amine oxidation with PLLA-LTv mats and TEMPO at 20%, and results are collected in Table 3, entries 1–10. The reaction conditions previously studied for free LTv [25,26] were adopted for the first attempt: amine (0.25 mmol) in acetate buffer pH = 4.5 (3 mL) and TEMPO 20% mol with respect to the amine. PLLA-LTv-A, PLLA-LTv-B, or PLLA-LTv-C were added to the solution and allowed to wet, then the solution was saturated with oxygen, and the closed vial was subjected to orbital shaking at 400 rpm at room temperature.

As the control experiment, a blank mat of PLLA (without LTv) and TEMPO 20% was used in the oxidation of 1a but it was ineffective, the conversion was poor, and the crude showed traces of a mixture of the expected aldehyde 2a together with imine 3a (Table 3, entry 1). On the contrary, PLLA-LTv-A, PLLA-LTv-B, and PLLA-LTv-C are all effective in giving 2a as the unique product, with PLLA-LTv-B as the best catalyst (Table 3, entries 2–4).

The use of the PLLA-LTv-A mat resulted in lower conversion and yields %, whereas PLLA-LTv-C gave a result similar to PLLA-LTv-B, in accordance with the actual amounts of LTv in mats and their activity (U) (Table 3, entries 2–4). With PLLA-LTv-B, different amounts of mat, substrate concentrations, and aqueous solvents were investigated.

The best result was achieved with 24.9 mg of PLLA-LTv-B (0.12 U) and 42 mM 1a in acetate buffer pH 4.5 (Table 3, entry 5), obtaining the aldehyde 2a in 79% of isolated yields. Interrupting the reaction at a lower reaction time, three days, and with a different work-up, the preferred formation of imine 3a was observed (Table 3, entry 6). As previously demonstrated in LTv oxidation of benzylamines [26], the imine formation was due to incomplete conversions that gave the condensation of the unreacted amine with the aldehyde as the early product. Concerning the oxidation of amines in aqueous solutions, the use of a buffer was strictly necessary because the amine dissociation in water increased the pH of the solution (around 10), leading to a complete inactivation of the laccase. Moreover, even a high buffer concentration could negatively affect the laccase activity.

Citrate gave results similar to acetate buffer, and a tentative to lower the acetate buffer concentration, or to use only one equivalent of acetic acid to form the ammonium acetate of 1a, gave worse results (Table 3, entries 8–10). Amines 1b-d were then tentatively investigated: 1b and 1c gave encouraging results (Table 3, entries 11 and 13), but the amine 1d, which has a secondary benzylic carbon atom, gave the corresponding *p*-OMe acetophenone in poor isolated yields (Table 3, entry 14). It could be possible that issues due to limited diffusion of substrate or product in the LTv mats occurred in this case.

The reuse of the PLLA-LTv-B mat was evaluated on the oxidation of amine 1a, under the conditions of entry 2 (Table 3), refreshing the same mat with a new buffer, TEMPO, and substrate (Figure 4). We observed a progressive decrease in product yields of 2a, reaching a residual efficiency of 22% after the fourth reuse (Figure 4A). Decreasing the concentration of acetate buffer to 0.2 M (data not shown) a significant decrease in 2a yields was obtained just from the second recycle, probably due to an improper buffering of the amine and a consequent deactivation of laccase at a more basic pH.

#### 3.4.2. Oxidation of Alcohols

Aerobic bio-oxidation of alcohols is one of the most challenging reactions for the synthesis of pharmaceuticals and fine chemicals to find more eco-sustainable oxidation of alcohols [37]. In a previous work, LTv resulted quite effectively in the two-step oxidation of primary alcohols to the corresponding carboxylic acids with TEMPO as mediator in a successful application for the synthesis of some bioactive anti-inflammatory agents belonging to the Profens’ class [25]. In the perspective to extend the use of the immobilized fungal laccase PLLA-LTv in alcohol oxidation, three alcohols were early selected for tentative oxidation reactions by PLLA-LTv-B and TEMPO: 2-phenyl propanol and ibuprofenol as models for the oxidation of profenols to the pharmacologically active profens, and 2-pyridine methanol for its excellent reactivity in the oxidation by LMS (Figure 1).

The optimized reaction conditions, previously described, were chosen and PLLA-LTv-B mats were tested in Milli-Q water TEMPO 20% with an alcohol concentration of 83 mM. Unfortunately, no substrate was oxidized, neither to the intermediate aldehydes nor to the carboxylic acids. Replacing Milli-Q water with acetate buffer at pH 4.5 or using co-solvents such as methanol or acetone to improve the solvation of the starting 4b alcohol did not give any improvement, instead it was observed that a damaging of mats occurred for prolonged reaction times, a phenomenon that was not evidenced in the oxidation of amines.

This negative behavior in the oxidation of alcohols by the immobilized laccase PLLA-LTv-B is in contrast with the previous results obtained by the use of the free enzyme LTv [25] and with the good results obtained with the same PLLA-LTv-B in the amine oxidation above reported.

The mechanism of TEMPO-mediated oxidation of alcohols and amines has been under continuous investigations [5]. Despite the radical nature of TEMPO, this mediator follows an ionic mechanism [38]. Two types of pre-oxidation intermediates were proposed (Figure 2): one derived from the attack of the alcohol on the nitrogen atom of the oxammonium cation, and the other that accounts for a hydride transfer [39,40]. In both cases a base-catalyzed hydrogen abstraction is involved to get the final oxidized product, as demonstrated by studies with organic bases in accelerating the reaction [41]. In the LTv-TEMPO oxidation of amines, the reactant amine could participate as a base to drive the reaction. In the case of alcohols, we observed that the free LTv and TEMPO are sufficient to obtain the carboxylic acids in Milli-Q water alone, whereas the immobilized PLLA-LTV with TEMPO in the same solvent conditions was not able to obtain the final oxidation product. This result could suggest a possible action of the enzyme even in the hydrogen abstraction to obtain the final oxidized product. The immobilization of the laccase could suppress this support action by a possible local conformational modification that could prevent the availability of a suitable basic residue necessary to the reaction. However, this early hypothesis must be supported by specific studies to come.

## 4. Conclusions

Laccase Tv was successfully encapsulated in PLLA to obtain PLLA-LTv nanofibers by a simple and reproducible electrospinning methodology. The PLLA-LTv mats with different amounts of LTv (PLLA-LTv-A, PLLA-LTv-B, and PLLA-LTv-C) were obtained and fully characterized. Release studies in Milli-Q water and buffer acetate pH = 4.5 revealed an increased tendency to release the enzyme on lowering the pH, probably due to a swelling effect of the polymer. Enzyme activity determination of PLLA-LTv samples and their application in LMS oxidation reactions allow us to evaluate the impact of different nominal amounts of laccase on the quality of the PLLA-LTv catalysts. The mat PLLA-LTv-B with 0.12 U in laccase showed the best encapsulation efficiency with a sufficient enzyme retention, and it worked successfully in the oxidation of benzylamines to the corresponding aldehydes in good conversions and yields. The immobilized enzyme PLLA-LTv was easily removed from the reaction batch and maintained a good enzyme stability to be reused in the next reaction cycles. Unexpectedly, PLLA-LTv mats were fully inactive in the two steps oxidation of alcohols thus showing their limitations. A tentative hypothesis on a direct action of the enzyme in the mechanism of the reaction could be formulated.

## Data Availability

Not applicable.

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
