# Peer review of "Laccase-Carrying Polylactic Acid Electrospun Fibers, Advantages and Limitations in Bio-Oxidation of Amines and Alcohols"

_jfb, 2022, doi:10.3390/jfb14010025_

Round 1

Reviewer 1 Report

The manuscript entitled “Laccase-carrying polylactic acid electrospun fibers, advantages and limitations in bio-oxidation of amines and alcohols.” describes the immobilization of Laccase from Trametes Versicolor (LTv) in poly-L-lactic acid (PLLA) nanofibers and its application in LMS oxidation reactions. There are some issues should be addressed:

1. The language and picture quality of the whole article should be carefully checked. For example:Line 83, “1,49U/mg” should be changed to “1.49U/mg”, in Figure S3, the line intersects the coordinate.

 2. Line 101, “The aqueous Laccase solution was then added to the PLLA organic solution”, Whether Laccase will be deactivated in organic solvents such as DMF.

 3. Please supplement and analyze the UV-Vis spectrum data, the FT-IR spectrum data, and NMR data where appropriate in the article, instead of using too many tables.

 4. Line 142, the value of λ is different from the supporting material. Please verify the λ value.

 5. Benzoquinone is an environmental pollutant. Is it appropriate to convert catechol to benzoquinone for wastewater treatment?

 6. Line 183, Please indicate the buffer solution of this system?

Author Response

Dear Editor,

Here the point-by-point response to the referee’s requests.

Reviewer 1

The manuscript entitled “Laccase-carrying polylactic acid electrospun fibers, advantages and limitations in bio-oxidation of amines and alcohols.” describes the immobilization of Laccase from Trametes Versicolor (LTv) in poly-L-lactic acid (PLLA) nanofibers and its application in LMS oxidation reactions. There are some issues should be addressed:

  1. The language and picture quality of the whole article should be carefully checked. For example:Line 83, “1,49U/mg” should be changed to “1.49U/mg”, in Figure S3, the line intersects the coordinate.

Line 83: Done

Figure S3: To see the smallest absorption peak it should be necessary to increase the spectrum intensity and doing so the most intense bands at 1100 nm of the O-C=O bending of the polyester PLLA goes out-of-scale, necessarily.

  1. Line 101, “The aqueous Laccase solution was then added to the PLLA organic solution”, Whether Laccase will be deactivated in organic solvents such as DMF.

Laccases and LMS showed to be tolerant of organic solvents (as a recent example of LMS in benzyl alcohol oxidation: Tetrahedron 2022, 128, 133114. The contact between Laccase and the DMF was limited to the electrospinning of the mat. As a matter-of-fact, the Laccase activity was evaluated on the mat with catechol, and it resulted active.

  1. Please supplement and analyze the UV-Vis spectrum data, the FT-IR spectrum data, and NMR data where appropriate in the article, instead of using too many tables.

Sorry, but we do not understand well this request which seems to be  not appropriate to our manuscript. In the body text or in the supplementary file there are not NMR, IR or UV Tables.

  1. Line 142, the value of λ is different from the supporting material. Please verify the λ value.

Values are correct, because the Comassie Blu has a maximum absorption at λmax = 595 nm, whereas we evaluated the protein amount at λ =620 because the fixed length selector of the multiscan instrument. We did changes in the supplementary file in the section “Bradford assay details”

  1. Benzoquinone is an environmental pollutant. Is it appropriate to convert catechol to benzoquinone for wastewater treatment?

Aqueous waste in paper industries was contaminated by lignin and polyphenols derivatives such as catechol or polyphenols, for instance. Treatment of those wastes with Laccase will result in the formation of polymeric products from quinones intermediates obtained by Laccase oxidation, which could be easily separated and eliminated from water by simple filtration.

  1. Line 183, Please indicate the buffer solution of this system?

In Table 3 for oxidation of amines, three buffer systems were used (acetate, citrate and H2O+AcOH), appropriate citation of Table 3 was added at line 198.

Reviewer 2 Report

Recommendation: Major revision

Comments:

1.      Please, check the title carefully. There should be no full stop in the title.

2.      The abstract and conclusions do not support the title. What are the advantages and limitations? How did the authors overcome the limitations?

3.      Line 232-233: why were the average diameters increased with the increase of laccase content? A proper explanation should be added here. Several factors govern the morphology of the nanofibers (for example https://doi.org/10.3390/pharmaceutics11070305).

4.      Can authors provide TGA curves of the prepared samples?

5.      The wettability of the materials should be reported. A water contact angle is required.

Author Response

Point-by point responce to referee

Reviewer 2

  1. Please, check the title carefully. There should be no full stop in the title.

Full stop eliminated

  1. The abstract and conclusions do not support the title. What are the advantages and limitations? How did the authors overcome the limitations?

Thank you for this comment. We added some sentences in the abstract and conclusions to clarify these points.

  1. Line 232-233: why were the average diameters increased with the increase of laccase content? A proper explanation should be added here. Several factors govern the morphology of the nanofibers (for example https://doi.org/10.3390/pharmaceutics11070305).

An increase of the fiber diameter is in accordance with literature data on immobilization of proteins in electrospun fibers, and it was attributed to an increase in the viscosity of the electrospinning solution. Reference 32 was added to support this effect.

  1. Can authors provide TGA curves of the prepared samples?

Thank you for this suggestion. We did the analyses and we added TGA curves in the Supporting Information and a short comment at line 259

  1. The wettability of the materials should be reported. A water contact angle is required.

Thank you for this suggestion. We did the analyses and we added contact angle measurements in the Supporting Information and a short comment at line 287.

Reviewer 3 Report

Dear authors,

I would like to thank you for sharing yours studies and results.

I have a few formal comments and recommendation.

- Please, briefly describe the need for oxidation of amines and alcohols in the introduction. In what specific application will it be used? 

- Please, check and make a correction in numbering of Figures in the text.

3. Results and Discussion, line 230 .... SEM are presented in Figure 1 and Figure S1. 

- in Figure S1 is Examples of calibration curves in water (A) and acetate buffer 0.1 M (B), SEM is in Figure S2.

- 3.3. Release of Laccase Tv .... depicted in Figure 2. But, DCS curves are in Figure 2, release of laccase is in Figure 3.

Besides, I must state that the manuscript is well readable and clear. The content is original and the results are interesting. Therefore, I recommend publishing after minor revision.

Author Response

Point-by-point responce to

Reviewer 3

- Please, briefly describe the need for oxidation of amines and alcohols in the introduction. In what specific application will it be used? 

The oxidative dehydrogenation reactions of amines and alcohols to aldehydes or acids, respectively, are functional group transformations commonly used in organic synthesis. In particular oxidation of amines or alcohols allows to obtain aldehydes or carboxylic acids, respectively. However, many of these systems still require transition metals or metal complexes as catalysts, harsh reaction conditions, and give metal-containing wastes. As discussed in the introduction, laccases are potential candidates in the quest for green alternatives to classic oxidants. A sentence was added at line 76.

- Please, check and make a correction in numbering of Figures in the text.

 Thank you very much for these corrections, we corrected both 3. and 3.3 with the right numbering

  1. Results and Discussion, line 230 .... SEM are presented in Figure 1 and Figure S1. 

- in Figure S1 is Examples of calibration curves in water (A) and acetate buffer 0.1 M (B), SEM is in Figure S2.

Thank you very much for this correction, done

- 3.3. Release of Laccase Tv .... depicted in Figure 2. But, DCS curves are in Figure 2, release of laccase is in Figure 3.

Thank you very much for this correction, done

Round 2

Reviewer 2 Report

The paper can be accepted now.